# The Effects of Active Video Game Exercise Based on Self-Determination Theory on Physical Fitness and Cognitive Function in Older Adults

**DOI:** 10.3390/jcm11143984

**Published:** 2022-07-08

**Authors:** Chenglei Zhao, Chenxi Zhao, Yunfeng Li, Minmin Zhao, Lin Wang, Jiawei Guo, Longhai Zhang, Yuliang Sun, Xintong Ye, Wenfei Zhu

**Affiliations:** School of Physical Education, Shaanxi Normal University, Xi’an 710119, China; 200823@snnu.edu.cn (C.Z.); zhaochenxi95@snnu.edu.cn (C.Z.); yunfenglee@snnu.edu.cn (Y.L.); zhaominmin@snnu.edu.cn (M.Z.); wletitia@snnu.edu.cn (L.W.); guojiawei729@snnu.edu.cn (J.G.); hae@snnu.edu.cn (L.Z.); ysun@snnu.edu.cn (Y.S.); ye@snnu.edu.cn (X.Y.)

**Keywords:** active video game, physical fitness, cognitive function, older adults

## Abstract

Background: Aging and physical inactivity are associated with declines in physical fitness and cognitive function. Active video games have proven to be beneficial for the physical health of older adults, but the exact effect of active video games on physical fitness and cognitive function was still unclear. Based on self-determination theory (SDT), which is a widely used theory of healthy behavior change, this study aimed to explore the effects of an active video game intervention on fitness and cognitive function in older adults. Methods: A total of 38 participants (mean age = 65.68 ± 3.78 years, 24 female) were randomly assigned to either an intervention group (active video game training) or a control group (no additional intervention). The participants in the intervention group trained for a total of 36 sessions (3 times per week for 50–55 min each) for 12 weeks. The control group continued with their normal daily living. The pre- and posttest measurements included: IPAQ-C score and physical fitness (BMI, body fat percent, blood pressure, reaction time, sit and reach, vital capacity, grip strength, static balance, blood biochemical tests for liver function, kidney function, blood lipids, glucose and insulin levels) and cognitive functions (processing speed, spatial ability, working memory, language ability, associative memory). Result: The intervention group showed a significantly smaller decrease in total average physical activity relative to the control group. BMI, vital capacity, systolic blood pressure, diastolic blood pressure, and spatial cognition significantly improved after training in the intervention group (BMI: *F* = 9.814, *p* = 0.004, d = −0.93, vital capacity: *F* = 4.708, *p* = 0.038, d = 0.67, systolic blood pressure: *F* = 5.28, *p* = 0.028, d = −0.68, diastolic blood pressure: *F* = 6.418, *p* = 0.016, d = −0.86, spatial cognition: *F* = 8.261, *p* = 0.007, d = 0.72). Three measures of static balance (closed eyes) also showed improvements after training (total length of swing: *F* = 3.728, d = −0.62, total velocity of swing: *F* = 3.740, d = −0.62, total area of swing: *F* = 2.920, d = −0.70). No significant training effects were evident in the results from the blood biochemical tests. Conclusion: This study indicates a positive influence of active video game training on physical fitness and cognitive function. The use of SDT-based active video game exercise as a feasible, safe, and effective training method for improving community older adults’ healthy, promoting group cohesion, and increasing motivation to exercise.

## 1. Introduction

The worldwide population of older adults is growing rapidly [1]. There are 26,736,000 people aged 60 or above in China, accounting for 18.9% of the national population [2]. Age-related deterioration includes poor balance and posture control, decreased reaction capacity, and physical weakness, as well as a decline in cognitive performance [3,4]. In addition, increased sedentary behavior has been associated with increased all-cause mortality and chronic disease among older adults, as well as an increased risk of falls [5]. These problems have caused great social and medical pressure and personal economic burden [6]. Numerous studies have shown that exercise can improve people’s health, prevent or delay age-related diseases, and improve the cognitive function of older adults [7,8,9]. However, with the outbreak of the COVID-19 pandemic, a large number of public sports venues closed, including stadiums, fitness centers, arenas, yoga rooms, and dance rooms. Exercise promotion became an essential trend during periods of home quarantine [10,11].

Self-determination theory (SDT) is a widely used theory of healthy behavior change that has been proven to be effective in improving the intrinsic motivation of adults to exercise [12]. The theory assumes that by fulfilling three basic human psychological needs, ability, autonomy, and relatedness, one’s motivation develops along a continuum from no motivation to external motivation and then to internal motivation, which helps to cultivate long-term exercise [12,13]. Many studies have shown that SDT can increase people’s motivation for physical activity, and a recent study found that an SDT-based, video-tailored physical activity intervention could improve participants’ physical activity [13,14]. In addition, studies have shown that participants were intrinsically motivated to participate in active video games because of the enjoyment experienced when playing the active video games and the perceived improvements in their physical and mental health and social confidence [15]. Thus, the influence of an active video game intervention based on SDT on the health of older adults is worth exploring.

As an electronic product for indoor activities of entertainment and fitness, active video games have been widely used in homes around the world. Active video games are a combination of sports and games where participants are physically active as they play. Studies comparing active video games with traditional exercise (treadmill, bicycle, etc.) have shown that adolescents and adults were more interested in active video games and that playing active video games elicited higher enjoyment [16,17]. In addition, studies have shown that active video games appear to be an interesting mode of home-based exercise for tackling anxiety disorders and sedentary behavior [18]. A large number of studies have shown that active video games can improve health-related conditions in older adults, with positive effects on balance, cognitive function, and physical function [19,20,21]. However, the effects of active video games on physical fitness in older adults are still being explored, and less evidence supports that active video games can adequately increase fitness levels and bring significant health benefits [22,23]. In addition, conventional active video game interventions did not emphasize intrinsic motivation for participants to keep physical activity, but an effective behavior change intervention should be grounded in a behavior change theory (such as SDT) that fosters more intrinsically regulated exercise motivation among participants. 

Therefore, this study’s purpose was to examine the effects of an active video game intervention based on SDT on older adults’ physical activity, physical fitness (BMI, body fat percent, blood pressure, reaction time, sitting forward, vital capacity, grip strength, static balance, blood biochemical tests for liver function and kidney function, blood lipids, glucose and insulin levels), and cognitive function compared with a health education control group. We hypothesized that compared with the control group, participants in the intervention group would show significantly improved physical fitness and cognitive function after 12 weeks of the active video game intervention. Findings from this study may establish a better system of SDT-based active video game interventions for improving older adults’ physical fitness and cognitive function to help attenuate the compounding effects of the COVID-19 pandemic on the physical inactivity and age-related healthy function decline in older adults.

## 2. Materials and Methods

### 2.1. Participants

Participants were recruited from communities through public advertisements in online publicity videos and offline presentations in Xian Province and Shaanxi Province, China, from July to September 2020. Participants fulfilling all of the following inclusion criteria were included in the study: (1) Age ≥ 60 years; (2) Having normal cognitive comprehension according to the Mini-Mental State Examination (MMSE) [24]; (3) Having healthy status through responses to a health survey, confirmations of disease history, and scores on the Physical Activity Readiness Questionnaire (PAR-Q) to ensure the objectivity and safety of the study [25]; (4) Possessing normal visual functions and being able to walk more than 10 min with or without the aid of an assistive device; and (5) Not having participated in similar intervention study within the last year. The exclusion criteria were as follows: (1) Cognitive dysfunction; (2) Diabetes mellitus and neuropathy or peripheral artery disease; (3) Dyskinesia induced by orthopedic or neurological diseases; (4) Other dyskinesia diseases that prevent training participation; (5) Having dropped out for personal or objective reasons resulting in the inability to complete the intervention or posttest. Written informed consent was obtained from all participants before this study began. The study was approved by the Shaanxi Normal University Ethics Committee (202016001).

### 2.2. Study Design and Procedure

This was a randomized controlled trial (RCT) that used an experimental mixed design with group (intervention, control) as a between-subject factor and time (pre, post) as a within-subject factor. The participants were randomly allocated to either the intervention group (IG) or the control group (CG) using computer-based random sorting and grouping, and each participant was numbered and then randomly assigned to one of the two groups using SPSS software.

The sample size for this study was calculated using G*Power software. ANOVA was used, and the average effect size d was set to be 0.5. When the significance level was 0.05, the power was 0.80, and the results showed that there should be at least 17 participants in each group. In order to ensure sufficient sample size after the intervention, the experiment recruited 24 subjects to be in the intervention group and 24 for the control group. We finally collected valid data for 38 participants (22 in the intervention group and 16 in the control group).

This study adopted non-blinded random grouping. All participants underwent pre-baseline testing within two weeks of the start of the study, and a posttest was conducted after the intervention phase. The test included physical activity, physical fitness (BMI, body fat percent, blood pressure, reaction time, sitting forward, vital capacity, grip strength, static balance, blood biochemical tests for liver function, kidney function, blood lipids, and glucose and insulin levels), and cognitive function. The participants in the intervention group trained 3 times per week for 50–55 min per session for 12 weeks for a total of 36 training sessions. The participants in the control group were advised to continue with their routine lives.

As shown in the flow chart (Figure 1), 56 individuals were initially screened for participation in this study. Among them, 48 qualified individuals were randomly assigned to either the intervention group (IG, 24 participants) or the control group (CG, 24 participants). Of the 48 participants, 10 did not complete the study due to various factors (2 in the intervention group and 8 in the control group). Therefore, a total of 38 participants (22 in the intervention group and 16 in the control group) completed all the interventions and assessments. None of the participants reported any adverse events.

### 2.3. Intervention

This study systematically combined SDT with an active video game intervention, aiming to improve participants’ motivational state from external to more internally regulated by targeting three basic psychological needs of participants:(1)Competence: Speak to the participants about the risks of low physical activity and the benefits of regular exercise; there are also professional researchers for professional training monitoring and guidance;(2)Autonomy: Visually demonstrate more and less intense exercise varieties to allow participants to choose which version to perform based on their level of physical fitness;(3)Relatedness: Conduct group scoring ranking and establish a reward system for persistence exercise.

In addition, for progressive changes in participants’ motivational state, the training session was divided into three stages:(1)The professional researchers conducted guided training throughout the course. The researchers instructed the participants in movements and techniques, at the same time leading the participants to exercise and encouraging them to increase their sense of achievement and enjoyment.(2)Participants were divided into groups and competed for scores between groups. These researchers were responsible for the monitoring game scores and establishing a scoring system and did not conduct the guided training.(3)Participants performed online video exercises at home, and the researchers conducted safety monitoring online. (The original plan of the study was that participants would train autonomously according to routine, with noninterference from researchers except for safety monitoring and start-up hardware devices. However, due to the local coronavirus virus outbreak, the local policy required residents to observe home quarantine to prevent the spread of the virus, so offline training was changed to online training at home. Online training did not violate the principles of the experimental design; that is, participants were allowed to train independently without interference).

This study presents the participants’ motivational states and external supervision intensity changes in Figure 2.

Training frequency and duration followed previous active video game studies illustrating current recommendations for physical activity and fall prevention in older adults [26]. Training was carried out at moderate to vigorous intensity, following similar studies in older adults [27,28]. Each session began with a warm-up and finished with a cool-down to reduce the risk of injury [29] During the training, each participant was required to wear a real-time heart rate monitor to ensure that participants’ heart rates were maintained at safe, moderate intensity levels [30]. In addition, before the experiment, we conducted a preliminary experiment on the participants. We let the participants try different games, and the final games were those with proper intensity and high levels of acceptance and enjoyment.

The intervention group included the following three games:(1)Zumba@ Fitness: Zumba Fitness is a dance game that consists of different music styles. For this study, each song was selected according to its intensity, rhythm, and acceptability to the participants. Each song lasted for 4 min, and there was a 1-min break after every 2 songs.(2)Fitboxing2 Aerobic boxing: This is a boxing virtual game. Participants held the handle to control their player. When the block entered the area, participants tried to punch toward the screen with the handle using different techniques such as hook, jab, dodge, dive dodge, U-bend, and step.(3)Mario Tennis Ace: This is a tennis virtual game that adopted the swing mode for this intervention. Participants held the handle as a racket and swung left or right to control the player to hit the ball. In the swing mode, the game provides an auxiliary movement function in which the player will semi-automatically move to the position of hitting, and two participants fight each other.

The hardware devices used in this group were a Switch (Nintendo Co., Ltd., Kyoto, Japan) and Joy-Con^TM^. The switch was used as the host machine and the Joy-Con as the handle. Participants wore or held the Joy-Con to control the player activities on the screen (HNNOR, LOK-360S, 3840 × 2160, Shenzhen Zhixin New Information Technology Co., Ltd., Shenzhen, China). The Fit Mao real-time heart rate monitor was also used (Fit Mao, Shandong Delay Cook Health Care Equipment Co., Ltd., Dezhou, China). The online video software used for supervising was Tencent Meeting (Shenzhen Tencent Computer System Co., Ltd., Shenzhen, China).

### 2.4. Outcome Measures

#### 2.4.1. Physical Activity Questionnaire

Physical activity was evaluated using the International Physical Activity Questionnaire Short Form (IPAQ-C), which has been documented to have both reliability and validity in physical activity environments [31,32]. The questionnaire referred to the last 7 days and asked about walking, moderate-intensity activities, vigorous-intensity activities, and sitting time.

#### 2.4.2. Physical Fitness Test

Body fat percent was measured with an InBody 230 Body Composition Analyzer (InBody Co., Ltd., Seoul, Korea). Systolic blood pressure and diastolic blood pressure were measured with an OMRON HEM-U30 electronic sphygmomanometer (OMRON Co., Ltd., Dalian, China). Vital capacity was measured with an HK6800-FH Electronic Spirometer (Shenzhen Hengkangjiaye Technology Co., Ltd., Shenzhen, China). Grip strength was measured with a CAMRY Electronic Hand Dynamometer (Zhongshan Camry Electronic Co., Ltd., Zhongshan, China). Height, weight, BMI, reaction time, and sit and reach were measured using an artificial intelligence fitness testing machine (iDong^@^TA106, Shenzhen Taishan Sports Technology Co., Ltd., Nanning, China).

The static balance data were obtained with a three-dimensional force platform (Kistler 9287CA, Winterthur, Switzerland). The force platform was fixed on the flat floor, and static balance was tested in the bipedal stance in two different conditions as below: (a) standing on the force platform with eyes open and (b) standing on the force platform with eyes closed. Measurement, Analysis and Reporting Software (MARS, version 2.1.0.8, S2P, science to practice, Ltd., Ljubljana, Slovenia) was used for comprehensive analyses of the force plate measurements. Sway path total length (mm) refers to the common length of the trajectory of the center of pressure (COP) sway calculated as a sum of the point-to-point Euclidian distances. Sway velocity total (mm/s) refers to the common length of the trajectory of the COP sway calculated as a sum of the point-to-point Euclidian distances divided by the measurement time. Sway area total (mm^2^) is defined as the total area swayed by the COP trajectory concerning the central stance point, calculated as the ellipse containing 95% of COP.

The blood biochemical tests were conducted at the same time of day (8:00–9:00 a.m.) for all participants. After an 8–12 h fast, intravenous blood sampling was performed on each participant at the hospital using the AU480 Automatic Biochemical Analyzer (Beckman Coulter Co., Ltd., Brea, CA, USA) followed centrifuging at 3000 rpm (TDL-5C, Shanghai Anting Scientific Instrument Factory, China) for 10 min. Total bilirubin, direct bilirubin, indirect bilirubin, glutamic–pyruvic transaminase, glutamic–oxalacetic transaminase, urea, creatinine, uric acid, total cholesterol, triglyceride, high-density lipoprotein cholesterol (HDL-C), low-density lipoprotein cholesterol (LDL-C), and glucose and insulin levels were measured. All assays were processed in a single batch at an accredited testing laboratory.

#### 2.4.3. Cognitive Testing

This study adopted the software Basic cognitive ability developed by Professor Li Deming from the Institute of Psychology, Chinese Academy of Sciences [33]. The testing software has been approved by the Professional Committee of Measurement of the Psychological Society and the Institute of Psychology of the Chinese Academy of Sciences. The selected tests for this study included the five dimensions of cognitive function: processing speed (symbol search), spatial ability (origami), working memory (operation span), language ability (analogy test), associative memory (portrait memory). Each question required participants to complete it within a certain amount of time, and higher scores indicated better cognitive functioning.

### 2.5. Statistical Analysis

Mean (M) and standard deviation (SD) were calculated for all variables. For the quantitative outcomes, normality and homogeneity of variance were tested (*p* > 0.05). For the baseline data, independent t-test or chi-square test was conducted to examine the differences between groups. Baseline adjusted analysis of covariance was used to determine whether the outcome variables (health fitness, cognitive, and static balance) varied between the intervention and control groups at posttest. Effect size (Cohen’s d) was calculated to determine the differences between groups from baseline to posttest. Missing data were filled with the mean. The magnitudes of effect sizes were categorized as none (0 ≤ Cohen’s d < 0.20), small (0.20 ≤ Cohen’s d < 0.50), medium (0.50 ≤ Cohen’s d < 0.80), or large (Cohen’s d ≥ 0.80). All analyses were carried out using the Statistical Package for the Social Sciences (SPSS version 16.0 software, IBM Corp., Armonk, NY, USA), The level of significance was set to α = 0.05.

## 3. Results

The demographic characteristics of the participants are presented in Table 1. There d (*p* < 0.05). In addition, there were no significant differences between the two groups regarding the outcome measures during the pre-intervention test (*p* < 0.05). All participants in the intervention group adhered to 100% of the total training sessions.

Figure 3 shows the before-and-after changes in physical activity time between the two groups, with the intervention group showing a small decrease in total average physical activity (*F* = 3.798, *p* = 0.059, d = 0.38) relative to the control group. The control group showed a significant decrease in moderate average physical activity (*p* = 0.022).

Table 2 presents the physical fitness, blood biochemical test, and static balance outcomes at posttest. For the physical fitness measures, there were large effects on BMI; it was reduced by 0.5 ± 0.82 kg/m^2^ (*F* = 9.814, *p* = 0.004, d = −0.93), a significant improvement. The changes were accompanied by significant increases in systolic blood pressure (9.88 ± 12.84 mmHg; *F* = 5.28, *p* = 0.028, d = −0.68) and diastolic blood pressure (6.5 ± 9.16 mm Hg, *F* = 6.418, *p* = 0.016, d = −0.85). The changes were accompanied by increases in vital capacity (191.36 ± 364.04 mL; *F* = 4.708, *p* = 0.038, d = 0.67), and significant improvement existed in vital capacity for the intervention group (*p* < 0.05). There were small improvement effects for grip strength, sitting forward, and reaction (d = 0.20–0.50) at the posttest, favoring the intervention. No substantive changes were observed in the body fat rate. For the blood biochemical tests, significant improvement existed in total cholesterol and low-density cholesterol for the control group (*p* < 0.05). There were small effects for direct bilirubin, uric acid, and triglyceride (d = 0.20–0.50) at the posttest, favoring the intervention. No substantive changes were observed in total bilirubin, indirect bilirubin, or glucose. For closed-eyes static balance, the changes were accompanied by improvements in total length of the swing (−40.6313 ± 118.29 mm; *F* = 3.728, d = −0.62), the total velocity of the swing (−2.023 ± 5.9103 mm/s, *F* = 3.74, d = −0.62), and the total area of the swing (−67.2605 ± 177.1394 mm*s; *F* = 2.92, d = −0.70), favoring the intervention. For open-eyes static balance, there were small effects for improvement in the total length of the swing, the total velocity of the swing, and the total area of swing (d = 0.20–0.50) at post-test, favoring the intervention.

Table 3 presents the cognitive function outcomes at posttest. Spatial cognition increased by 0.86 ± 2.81 (*F* = 8.261, *p* = 0.007, d = 0.72) in the intervention group, a significant change. There were small effects for operation span and portrait memory (d = 0.20–0.50) at the post-test, favoring the intervention. No substantive changes were observed in the symbol search.

## 4. Discussion

In this study, we investigated the impacts of the SDT-based active video game training on motivation for physical activity, physical fitness, and cognition in older adults. After 12 weeks of active video game training, the results showed improvements in participants’ motivation for physical activity, physical fitness (BMI, blood pressure, vital capacity, static balance), and cognition (spatial cognition) compared with the control group. No training-related improvements were evident in the results for glucose and insulin levels, blood lipids, liver function, kidney tests, body fat rate, grip strength, sitting forward, or reaction time. This study may establish a better system of SDT-based active video game interventions for improving older adults’ physical fitness and cognitive function to help attenuate the compounding effects of physical inactivity and age-related healthy function decline.

This study methodically combined SDT with an active video game intervention, which aimed to progress participants’ motivational states from less to more internally regulated and maintain physical activity. We found that the intervention group participants were able to maintain physical activity even after the 12-week intervention was over. A recent study found overall negative impacts of COVID-19 and the winter season on physical activity [34,35], and our study’s result showed that participants in the control group showed a decrease in total average physical activity. However, this study’s distinct result indicated that SDT-based active video games motivated older adults to effectively maintain their physical activity in the intervention group during home quarantine in response to COVID-19 even with less external supervision. This study’s findings are similar to results from previous studies [13] that SDT-based exercise interventions are positive for people’s physical activity. This further confirms that SDT-based exercise interventions designed to increase people’s physical activity during the COVID-19 pandemic are feasible and effective [36].

A significant improvement was found in vital capacity following the active video game intervention compared with the control group after the 12-week intervention. In the present study, respiratory function improved through the active video game intervention. Research has reported that children had significant improvements in lung function after a 12-week active video game intervention [37]. In addition, several studies have confirmed that exercise can improve lung function and physical ability in adolescence and middle age [38,39,40]. The improved vital capacity in the intervention group proved that vital capacity can be improved in a short period (12 weeks) through intensive interventions.

The present study found consistent results with those of Tzu-Cheng Yu et al., who reported no change in grip strength or body fat percentage after 10 weeks of a Kinect intervention in healthy older adults [22]. However, a previous study presented different findings in which grip strength increased significantly in older adult women after 8 weeks of the Kinect intervention [41]. The discrepancy may be due to the strong male participants included in this study; for them, this relatively low-intensity exercise was not enough to improve their strength.

Three measures of static balance (closed eyes) showed improvements after training in favor of the control group. Studies have indicated improvement effects of active video games on static balance [42]. The reason there is no significant effect on other measures (open eyes) can be explained by the differences in equipment used for the measurements. Previous studies used the Berg Balance Scale [43], while balance ability was evaluated using a force plate system in this study. In addition, due to the change in ambient temperature, participants took the pretest in summer and the posttest in winter, and their clothes and their coldness might have affected the amplitude of swing during those tests.

On the physical fitness measures, BMI revealed a significant improvement after the 12-week active video game intervention training compared with the control group. This result was similar to those reported by Han-Chung Huang et al. that significant improvement was observed in BMI after 12 weeks of Xbox 360 training in healthy adults, relative to control [44]. However, some studies with an active video game intervention and fitness tests found no change in BMI compared with the control group [22]. The inconsistent results regarding the respective outcomes may be associated with considerable differences in study designs, enrolled participants, and active video games applied [22]. We would argue that the positive effects in this study are attributed to the active video game intervention based on SDT. 

Previously published studies have shown that exercise training can decrease high blood pressure [45,46], and there have also been studies showing that active video games can normalize blood pressure and keep it stable [44]. Our results in this research were the same as those from previous studies: the diastolic and systolic blood pressure in the intervention group were significantly better compared to the intervention group. Active video games can effectively control blood pressure deterioration in older adults. Blood pressure is an important indicator of cardiovascular disease (CVD), and high blood pressure increases CVD risk [47]. Therefore, the 12-week active video game intervention training can help prevent CVD and reduce the risk of CVD. Regular active video game training can maintain good elasticity in small arteries to offset weakening with senescence.

Significant improvements were found in spatial cognition in the intervention group after 12 weeks of active video game training. Many previous studies have reported the beneficial effects of active video games on cognitive function [20,48]. Stanmore et al. showed that active video games were beneficial for executive functioning, specifically for inhibitory control and cognitive flexibility [48]. However, a summary of several studies found that some reported improvements in executive functioning with active video games, while other studies failed to find that same effect [9]. The reason for the mixed results might be that similar active video game systems were superimposed were implemented under intervention designs and training characteristics that varied considerably across studies [9]. Maillot et al. demonstrated the significant benefits of active video game training on executive control and processing speed tasks, but there was no transfer of training on visuospatial measures [49]. This is a deviation from what we found, and we posit that the difference is due to different game projects and different ways of measuring. Guimarães et al. also indicated an improvement in executive function and delayed recall in the intervention group, and the cognitive benefits associated with the use of active video games may be related to visual stimuli and the characteristics of the game [50]. Consequently, further research is required to validate the specific physiological reasons for the cognitive improvements from specific active video game events.

Previously published studies have shown that exercise training can decrease triglycerides, glucose, and total cholesterol and reduce the risk of various diseases, such as hypertension, cardiovascular diseases, and diabetes [45,46,47,51,52]. However, there are few studies on the effects of an active video game intervention on the results of blood biochemical tests. In this study, no significant changes were detected in the intervention group compared with the control group. We found that about 58% of the participants in our study had chronic diseases, and another 42% were taking some medications, and medication may greatly affect the blood. In addition, due to the large individual differences between groups, we could not interfere with participants’ dietary habits and exercise habits. Therefore, the effects of active video games on blood lipids, glucose and insulin levels, liver function, and kidney function in older adults need to be further explored. However, this study has great reference value for the future exploration of the influence of an active video game intervention on blood biochemical testing results.

This study has strengths and limitations. The strengths include the following: (1) This study was a randomized intervention study with a control group. (2) The intervention was based on SDT and active video games and lasted for 12 weeks. Training intensity was monitored throughout the course to ensure the quality and safety of training. (3) The tests used objective measures to evaluate cognitive function, physical fitness, and blood sample. Its limitations include the following: (1) The sample size of the study was relatively small, and the loss rate was largely due to the COVID-19 epidemic. It would be more innovative to have three arms (active video game intervention, traditional exercise intervention, and control) to better understand the effects of active video games in older adults. Furthermore, the pandemic affected the results, so these conclusions only apply to this particular natural setting. (2) In this study, the pre- and posttest could have reflect different situations. However, as both the intervention and control groups experienced the same COVID-19 pandemic, the differences in the changes between these two groups can still be used to explore the effects of the active video game on physical fitness and cognitive function in this population in this natural setting. (3) We did not control for participants’ dietary habits or medication intake, so that we could not know the main factors in the participants’ blood changes. We also did not survey participants’ motivation for physical activity in SDT-based active video games. In the future, it is necessary to add other nutrition items and more participants (no chronic diseases) in future studies to have a more comprehensive perspective of the effects of active video game training on physical fitness in healthy older adults.

## 5. Conclusions

The findings of this study indicated that active video games based on SDT have a good effect on physical activity in older adults after 12 weeks of intervention during the COVID-19 pandemic. The older adults in the intervention group exhibited significant improvement in physical fitness (BMI, blood pressure, and lung capacity) and cognitive function (spatial cognition). Additional benefits observed in this study also suggested the use of SDT-based active video game exercise as a feasible, safe, and effective training method for improving community older adults’ health, promoting group cohesion, and increasing motivation to exercise. It may help to establish better SDT-based active video game interventions for improving older adults’ physical fitness and cognitive function to help attenuate the compounding effects of a sedentary lifestyle and age-related healthy function decline in older adults.

## Figures and Tables

**Figure 1 jcm-11-03984-f001:**
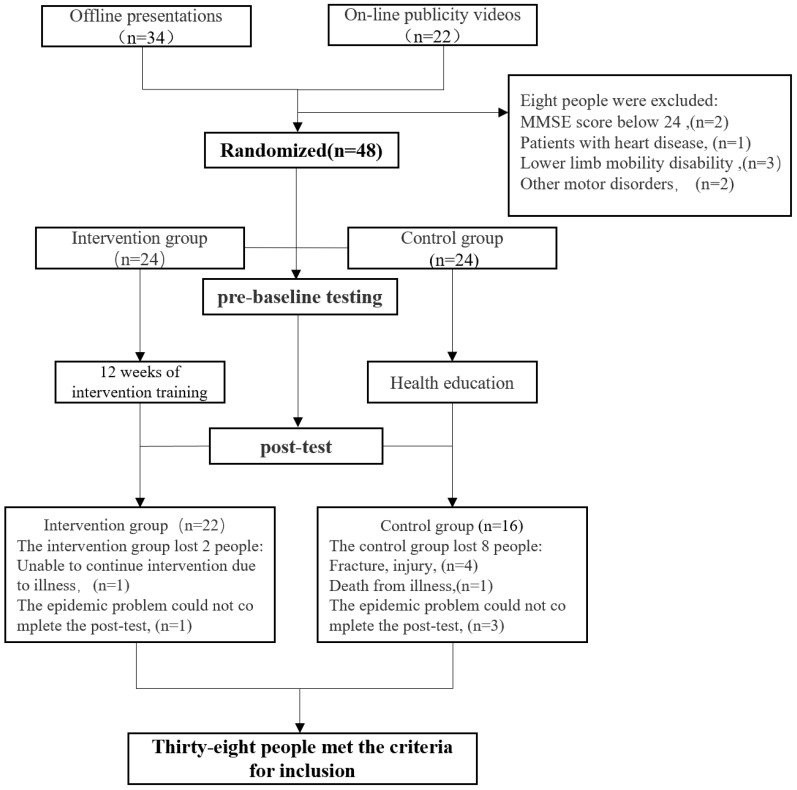
Study flow chart; MMSE: Mini-Mental State Examination.

**Figure 2 jcm-11-03984-f002:**
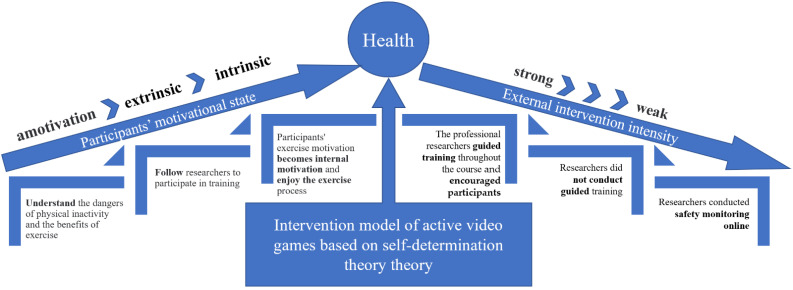
Motivational states and external intervention intensity changes.

**Figure 3 jcm-11-03984-f003:**
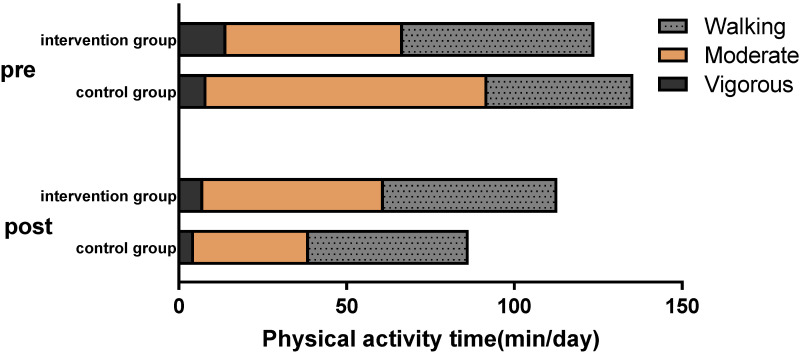
Pre- and post-intervention physical activity (including walking and moderate-to-vigorous physical activity, *F* = 3.798, *p* = 0.059, d = 0.38).

**Table 1 jcm-11-03984-t001:** Baseline demographic characteristics and screening values of the analyzed participants.

Index		IG	CG	Total	t/χ^2^	*p*
Number of people		22	16	38		
Age, M (SD)		65.64 (4.2)	65.75 (3.24)	65.68 (3.78)	−0.90	0.929
Female (%)		14 (63.6)	10 (62.5)	24 (63.20)	0.01	0.943
Mini mental screener examination, M (SD)		29 (1)	28.69 (1.1)	28.87 (1.07)	0.89	0.381
Education (%)					5.80	0.215
	Primary school	1 (4.5)	1 (16.3)	2 (5.3)		
	Junior high school	8 (36.4)	2 (12.5)	10 (26.3)		
	High school	5 (22.7)	7 (43.8)	12 (31.6)		
	college	6 (27.3)	2 (12.5)	8 (21.1)		
	Undergraduate course	2 (9.1)	5 (25)	6 (15.8)		
Smoking (%)		1 (4.5)	1 (6.3)	2 (5.3)	0.05	0.816
Alcohol consumption (%)		4 (18.2)	1 (6.3)	5 (13.2)	1.15	0.283
Industry engaged (%)					1.72	0.887
	professional	6 (27.3)	4 (25)	10 (26.3)		
	industrial	1 (4.5)	2 (12.5)	3 (7.9)		
	administrative	2 (9.1)	1 (06.3)	3 (7.9)		
	Agriculture	2 (9.1)	1 (06.3)	3 (7.9)		
	general	1 (4.5)	0 (00.0)	1 (2.6)		
	other	10 (45.5)	8 (50.0)	18 (47.4)		
Sports injury (%)					6.00	0.199
	knee	3 (13.6)	7 (43.8)	10 (26.3)		
	vertebral	5 (22.7)	1 (6.3)	6 (15.8)		
	diverse	1 (4.5)	0	1 (2.6)		
	other	3 (13.6)	1 (6.3)	4 (10.5)		
	without	10 (45.5)	7 (43.8)	17 (44.7)		
Medication (%)		11 (50)	5 (31.3)	16 (42.1)	1.34	0.248
Chronic diseases (%)					0.48	0.923
	One of the three tenors	4 (18.2)	4 (25.0)	8 (21.1)		
	diverse	4 (18.3)	2 (12.5)	6 (15.8)		
	other	5 (22.7)	3 (18.8)	8 (21.1)		
	without	9 (40.9)	7 (43.8)	16 (42.1)		
Vision and hearing problems (%)					8.60	0.072
	myopia	4 (18.2)	0	4 (10.5)		
	presbyopia	2 (9.1)	4 (25.5)	6 (15.8)		
	diverse	0	2 (12.5)	2 (5.3)		
	other	2 (9.1)	0	2 (5.3)		
	without	14 (63.6)	10 (62.5)	24 (63.2)		
Walking (min/day)		57.0 (44.4)	43.5 (41.1)	51.3 (43.0)	0.95	0.347
Moderate physical activity (min/day)		52.6 (88.9)	83.7 (83.9)	65.8 (87.1)	−1.09	0.283
Vigorous physical activity (min/day)		14.2 (31.7)	8.2 (16.7)	11.7 (26.3)	0.68	0.499
Total physical activity (min/day)		123.8 (97.2)	135.5 (103.1)	128.7 (98.5)	−0.36	0.724

Notes: IG: intervention group; CG: control group; SD: Standard deviation.

**Table 2 jcm-11-03984-t002:** Baseline (0 weeks), post-test (12 weeks) change for physical fitness variables.

	IG (*n* = 22), Mean (SD)				CG (*n* = 16), Mean (SD)				0 Weeks vs. 12 Weeks			
Index	0 Weeks		12 Weeks		0 Weeks		12 Weeks		F	*p*	d	Effect Size
**Physical fitness**												
BMI, kg/m^2^	23.84	(2.8)	23.33	(2.58)	23.65	(3.69)	23.86	(3.54)	9.814	0.004 *	−0.93	L
Body fat rate, %	24.78	(8.57)	25.95	(7.16)	25.13	(6.75)	26.14	(6.78)	0.070	0.793	0.03	-
Systolic blood pressure, mmhg	126.18	(15.13)	127.23	(16.14)	122.30	(12.90)	132.10	(8.90)	5.280	0.028 *	−0.68	M
Diastolic blood pressure, mmhg	75.68	(8.81)	74.91	(6.89)	70.80	(10.40)	77.30	(8.20)	6.418	0.016 *	−0.86	L
Vital capacity, mL	2570.82	(785.61)	2762.18	(991.1)	2646.60	(736.4)	2556.70	(844.9)	4.708	0.038 *	0.67	M
Grip strength, kg	28.31	(8.20)	27.76	(9.04)	30.30	(9.1)	28.70	(9.0)	0.647	0.427	0.39	S
Sitting forward, cm	5.68	(11.94)	5.96	(11.13)	5.20	(11.0)	6.90	(10.5)	0.932	0.342	−0.32	S
Reaction time, s	0.56	(0.09)	0.58	(0.07)	0.60	(0.1)	0.60	(0.1)	0.099	0.756	0.25	S
**Static balance**												
Total length of the swing, mm (open Eyes)	209.83	(51.81)	205.12	(60.44)	213.42	(47.65)	226.12	(49.62)	1.237	0.274	−0.22	S
Total velocity of the swing, mm/s (open Eyes)	10.49	(2.59)	10.26	(3.02)	10.67	(2.38)	11.31	(2.48)	1.231	0.275	−0.25	S
Total area of swing, mm × s (open Eyes)	212.90	(124.10)	181.63	(105.58)	194.34	(65.61)	202.78	(69.23)	1.097	0.303	−0.33	S
Total length of the swing, mm (Closed Eyes)	271.48	(82.40)	230.85	(74.57)	257.31	(80.44)	288.58	(95.86)	3.728	0.062	−0.62	M
Total velocity of the swing, mm/s (Closed Eyes)	13.57	(4.12)	11.55	(3.73)	12.87	(4.02)	14.43	(4.77)	3.740	0.062	−0.62	M
Total area of swing, mm × s (Closed Eyes)	271.18	(141.13)	203.92	(116.03)	216.80	(93.30)	269.10	(142.60)	2.920	0.097	−0.70	M
**Blood biochemical tests**												
Total bilirubin, μmol/L	16.87	(3.54)	13.42	(2.22)	16.29	(3.73)	13.82	(3.54)	0.666	0.421	−0.31	S
Direct bilirubin, μmol/L	7.02	(1.52)	3.94	(1.12)	6.84	(1.25)	4.30	(1.56)	1.632	0.211	−0.36	S
Indirect bilirubin, μmol/L	10.17	(2.63)	9.55	(1.88)	9.47	(2.66)	9.53	(2.26)	0.182	0.673	−0.30	S
Glutamic-pyruvic transaminase, μ/L	24.41	(18.86)	25.79	(13.25)	27.78	(33.88)	26.00	(12.78)	0.261	0.613	0.16	-
Glutamic oxalacetic transaminase, μ/L	22.85	(11.48)	21.96	(5.65)	19.47	(4.82)	22.22	(5.02)	0.075	0.786	−0.37	S
Glucose, mmol/L	4.80	(0.75)	5.15	(0.85)	4.87	(0.60)	5.14	(0.94)	0.500	0.485	0.14	-
Urea, mmol/L	5.33	(1.28)	6.74	(1.58)	5.55	(1.24)	6.38	(1.27)	2.780	0.105	0.51	M
Creatinine, μmol/L	69.92	(14.95)	73.36	(12.61)	72.80	(10.84)	73.56	(11.46)	1.062	0.310	0.33	S
Uric acid, μmol/L	331.86	(97.26)	279.55	(73.84)	346.83	(80.59)	309.19	(69.49)	1.691	0.203	−0.29	S
Total cholesterol, mmol/L	5.50	(0.96)	5.40	(0.97)	5.30	(1.17)	4.94	(0.91)	7.020	0.012	0.37	S
Triglyceride, mmol/L	1.39	(0.58)	1.49	(0.73)	1.43	(0.68)	1.56	(0.81)	0.082	0.777	−0.05	-
High density cholesterol, mmol/L	1.59	(0.38)	1.51	(0.38)	1.62	(0.34)	1.43	(0.35)	1.964	0.171	0.40	S
Low density cholesterol, mmol/L	3.35	(0.73)	3.23	(0.78)	3.10	(0.79)	2.78	(0.53)	9.469	0.004	0.34	S
Insulin, μ/mL	6.69	(3.51)	5.87	(2.93)	6.68	(2.37)	6.19	(2.12)	0.001	0.977	−0.14	-

Note: IG: intervention group; CG: control group; BMI: Body Mass Index; SD: Standard deviation; *: *p* < 0.05; The magnitudes of the effect sizes were categorized as N (none): (0 ≤ Cohen’s d < 0.20), S (small): (0.20 ≤ Cohen’s d < 0.50), M (medium): (0.50 ≤ Cohen’s d < 0.80), or L (large): (Cohen’s d ≥ 0.80).

**Table 3 jcm-11-03984-t003:** Changes from baseline (0 weeks) to posttest (12 weeks) in the cognitive function variables.

	IG (*n* = 22), Mean (SD)				CG (*n* = 16), Mean (SD)				0 Weeks vs. 12 Weeks			
Index	0 Weeks		12 Weeks		0 Weeks		12 Weeks		F	*p*	d	Effect Size
Symbol search	29.14	(6.33)	29.91	(7.00)	30.10	(7.30)	30.00	(7.20)	0.103	0.751	0.19	-
Operations span	3.49	(2.00)	4.09	(4.09)	5.10	(3.20)	5.10	(3.10)	0.050	0.825	0.25	S
Portrait memory	19.50	(6.06)	19.59	(7.58)	20.70	(9.20)	18.20	(8.80)	1.413	0.243	0.38	S
Spatial cognition	5.00	(3.04)	5.86	(2.80)	4.90	(2.30)	3.90	(2.90)	8.261	0.007 *	0.72	M
Similar tests	33.41	(12.09)	32.91	(14.31)	31.40	(14.50)	36.00	(11.70)	0.847	0.364	−0.46	S

Note: IG: intervention group; CG: control group; SD: Standard deviation; *: *p* < 0.05; The magnitudes of the effect sizes were categorized as N (none): (0 ≤ Cohen’s d < 0.20), S (small): (0.20 ≤ Cohen’s d < 0.50), M (medium): (0.50 ≤ Cohen’s d < 0.80), or L (large): (Cohen’s d ≥ 0.80).

## Data Availability

The data presented in this study are available on request from the corresponding author. The data are not publicly available due to confidentiality.

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
