# Peer review of "The Effects of Active Video Game Exercise Based on Self-Determination Theory on Physical Fitness and Cognitive Function in Older Adults"

_jcm, 2022, doi:10.3390/jcm11143984_

Round 1
Reviewer 1 Report
I would like to thank the editor for the opportunity to review the manuscript entitled "Effect of active video games exercise on physical fitness and cognitive function in older adults".
This manuscript describes an RCT study about the effects of active video games on physical, cognitive, and hematological parameters in older adults based on the Self-determination theory. The topic is interesting as promoting a healthy and active lifestyle in this population is particularly important. The topic is not very novel, as several systematic reviews and meta-analyses have been published in the last years (e.g., Vazquez et al., 2018). As such, the novelty of this study is the implementation of SDT in the proposed active video games protocol. Therefore, I suggest highlighting this in the title.
Reviewer 2 Report
Dear authors,
I carefully read your manuscript, and I have some doubts or suggestions, which, I hope, will improve the quality of your article.
My major concerns are
1. the arms of the randomization since, in my opinion, it would have been more interesting and innovative to have three arms (video game, physical activity, control groups)
2. the small sample size
3. it seems that pre-evaluations have been conducted before the covid breakout and the post-test after or during covid isolation. Thus, pre and post-test could have shown different situations, as it is possible to see in figure 3, where the physical activity practice even decreased during the intervention study protocol.
Please, see later for minor comments:
1. It is unclear if the aim and the scope of the study changed due to covid breakout, and all the study has been revised to adapt to the covid situation.
2. In the introduction, among citations number 7 and 8, I suggest adding the following article: DOI: 10.1007/s11332-019-00573-x.
3. In line 102, you say you checked for the health status. Does this mean that the participants could have different physical activity levels? Did you adjust for this variable in the analysis?
4. Line 116, what does it mean “phase III study?”
5. Line 121: Why has the pre-test been done two weeks after the study? This could have negatively affected the results.
6. Post-test is written in different ways throughout the text.
7. 50-55 minutes are per week or per session?
8. From lines 177 to 198: are these validated methods?
9. Figure 3:
a. what does pro mean?
b. Are these differences significant?
10. High cholesterol?
11. LDL significantly changed, but no comments are present on this result.
12. Table 3: why did you add a and b?
13. Lines 325-325: physical activity decreased during the intervention.
14. Line 353: could the winter season have influenced the decrease in the amount of physical activity?
Reviewer 3 Report
I applaud the authors for this work. The manuscript presents a study about Effect of active video games exercise on physical fitness and cognitive function in older adults. The theme is relevant for current discussions on the use of video games and technologies in older adults.
· “Active video games, is one of the strategies used in recent studies to improve adherence to physical activity… I recommend explaining better what the definition of Active video games …… …… It is relevant to list the name and gender of games used (sports, adventure etc.).
· Describe how the participants were randomized
· Please add more information about the intervention (intensity, time of practice…. Day/week).
· How much power does this sample size have to estimate these changes?
· Add more information about biochemical analyze
· In the last section, conclusion, I would recommend the inclusion of the main contributions the study brought as possible reflections about the study findings for the area of health and video games in older adults.
Round 2
Reviewer 2 Report
Dear authors,
I appreciated the improvements after the revision process. I have some more and little points to specify better:
1. I suggest stressing the explanation and discussion of Figure 3 better since it represents the novelty of the study, in my opinion;
2. in Table 2, letters a and b are superfluous; furthermore, you used also the c letter without explaining it in the table description;
3. the suggested citation (DOI: 10.1007/s11332-019-00573-x, https://link.springer.com/article/10.1007/s11332-019-00573-x) is about the relation between sleep and physical activity in older adults, and it is not that added in the revised manuscript.
